# Association of *CTLA4* Gene Polymorphism with Transfusion Reaction after Infusion of Leukoreduced Blood Component

**DOI:** 10.3390/jcm8111961

**Published:** 2019-11-13

**Authors:** Ying-Hao Wen, Wei-Tzu Lin, Wei-Ting Wang, Tzong-Shi Chiueh, Ding-Ping Chen

**Affiliations:** 1Department of Laboratory Medicine, Linkou Chang Gung Memorial Hospital, Taoyuan 33305, Taiwan; b9209011@cgmh.org.tw (Y.-H.W.); berry0908@cgmh.org.tw (W.-T.L.); s1223@adm.cgmh.org.tw (W.-T.W.); drche0523@cloud.cgmh.org.tw (T.-S.C.); 2Graduate Institute of Clinical Medical Sciences, College of Medicine, Chang Gung University, Taoyuan 33302, Taiwan; 3Graduate Institute of Biomedical Sciences, College of Medicine, Chang Gung University, Taoyuan 33302, Taiwan; 4Department of Medical Biotechnology and Laboratory Science, Chang Gung University, Taoyuan 33302, Taiwan

**Keywords:** transfusion reaction, CTLA4, gene polymorphism, leukoreduction

## Abstract

Leukocytes and cytokines in blood units have been known to be involved in febrile non-hemolytic transfusion reaction (FNHTR), and these adverse reactions still occur while using pre-storage leukoreduced blood products. Blood transfusion is similar to transplantation because both implant allogeneic cells or organs into the recipient. CTLA4 gene polymorphism was found to be associated with graft-versus-host disease in hematopoietic stem cell transplantation. We performed a prospective cohort study at a major tertiary care center to investigate the correlation of CTLA4 gene polymorphism and transfusion reactions. Selected CTLA4 gene SNPs were genotyped and compared between patients with transfusion-associated adverse reactions (TAARs) and healthy controls. Nineteen patients and 20 healthy subjects were enrolled. There were 4 SNPs showing differences in allele frequency between patients and controls, and the frequency of “A” allele of rs4553808, “G” allele of rs62182595, “G” allele of rs16840252, and “C” allele of rs5742909 were significantly higher in patients than in controls. Moreover, these alleles also showed significantly higher risk of TAARs (OR = 2.357, 95%CI: 1.584–3.508, *p* = 0.02; OR = 2.357, 95%CI: 1.584–3.508, *p* = 0.02; OR = 2.462, 95%CI: 1.619–3.742, *p* = 0.008; OR = 2.357, 95%CI: 1.584–3.508, *p* = 0.02; OR = 2.357, 95%CI: 1.584–3.508, *p* = 0.02, respectively). The present study demonstrated the correlation of CTLA4 gene polymorphism and transfusion reaction, and alleles of 4 CTLA4 SNPs with an increased risk of TAARs were found. It is important to explore the potential immune regulatory mechanism affected by SNPs of costimulatory molecules, and it could predict transfusion reaction occurrence and guide preventive actions.

## 1. Introduction

Blood transfusion is an urgent intervention executed to restore lost components of blood on a short-term basis [1]. Transfusion reactions are also called transfusion-associated adverse reactions (TAARs), which are defined as adverse events associated with blood transfusion with one of the components of blood [2]. Acute transfusion reactions are distinguished from delayed transfusion reactions by occurring within 24 h after transfusion. The adverse reactions commonly occur in hospital, and the symptoms include fever, chills, rigors, dyspnea, hypotension, itching, and urticaria. They may range in severity from mild to life-threatening [2,3].

The causes of TAARs were suggested according to individual reactions [4]. Leukocytes and cytokines in blood units have been known to be involved in febrile non-hemolytic transfusion reaction (FNHTR) [5]. Human leukocyte antigen (HLA) antibodies would be induced in recipients if the HLA of transfused products was different from those in recipients, and FNHTR would occur when the same HLA antigens are transfused next time [6]. Therefore, these transfusion reactions commonly occurred in multiparous women [7] and multiply transfused patients [8].

Leukocyte-poor blood components were used to avoid alloimmunization to HLA antigens for recipients with a history of FNHTR and avoid being infected by cytomegalovirus (CMV) [9]. Moreover, the development of FNHTR was gradually decreased by using pre-storage leukoreduced blood components, because of less accumulation of cytokines in the blood components [10]. Van de Watering LM et al. proposed that leukocyte-depleted blood transfusion is beneficial for postoperative complications in patients undergoing cardiac surgery [11]. K Rajesh et al. indicated that execution of pre-storage leukoreduction significantly decreased FNHTR occurrence [12]. However, FNHTR still presents when transfusing pre-storage leukoreduced blood components. According to a previous study, FNHTR would occur when human platelet antigen (HPA) was unmatched between donor and recipient, and HPA-2 was related to FNHTR [13].

Hematopoietic stem cells (HSCs) are precursors that have the ability to differentiate into all types of blood cells, including leukocytes, erythrocytes, and platelets, which are found in blood products [14]. In 2009, Li XC et al. pointed out that costimulatory molecules play an important role in acute graft-versus-host disease (GVHD) [15] and it was found that polymorphism of costimulatory molecules, including CTLA4 and ICOS, might be associated with GVHD in allogeneic hematopoietic stem cell transplantation (HSCT) [16]. Moreover, both CTLA4 and ICOS gene polymorphisms in the donor and recipient might be of importance for the outcome of allogeneic HSCT [17]. In addition, there were several studies on HSCT that showed that CTLA4 SNPs have been associated with differences in relapse free survival (RFS), overall survival (OS), and GVHD, but there were discordant results between these investigators [18,19,20,21,22].

Blood transfusion is similar to transplantation in that it implants allogeneic cells or organs into a recipient. In addition, GVHD after transplantation results from the immune response, the same as TAARs after transfusion. Herein, we investigated the correlation of CTLA4 gene polymorphism and transfusion reactions, in order to uncover potential immune regulation affected by costimulatory molecules which could predict transfusion reaction occurrence and guide preventive actions.

## 2. Experimental Section

### 2.1. Study Subjects

This study was approved by the Institutional Review Board of Chang Gung Memorial Hospital (CGMH) with the approval ID of 1809100035. Nineteen patients with TAARs after leukoreduced blood products (leukocyte-poor RBC or leukocyte-poor platelet) transfusion at Linkou CGMH and 20 healthy subjects were included in the study. Investigation of TAARs was performed by clinical pathologists in Department of Laboratory Medicine at Linkou CGMH, and diagnostic criteria of TAARs were according to Hemovigilance Module Surveillance Protocol of National Healthcare Safety Network (NHSN) Biovigilance Component [23].

### 2.2. DNA Extraction

Peripheral blood samples were collected in EDTA-coated vacuum tubes, and genomic DNA was extracted by using QIAamp DNA Mini kit (Qiagen GmbH, Hilden, Germany) according to the manufacturer’s instructions. DNA concentration and purity were evaluated by measuring the optical density at 260 and 280 nm through a UV spectrometer.

### 2.3. PCR Amplification

The PCR mixture contained 1 µL DNA, 10 µL Hotstar Taq DNA Polymerase (Qiagen GmbH, Hilden, Germany), 1 µL CTLA4 forward primer (10 Mµ), 1 µL CTLA4 reverse primer (10 Mµ), and 12 µL ddH2O. The CTLA4 primers were shown in Table 1. The PCR program was 1 cycle of 95 ℃ for 10 min, 35 cycles of 94 ℃ for 30 secs, 65.5 ℃ for 30 secs, and 72 ℃ for 1 min. The final elongation step was 3 min at 72 ℃ and then soaking at 10 ℃. For gel electrophoresis visualization, 5 μL of the PCR products was pipetted onto a 1.5% agarose gel and run at 100 V for 20 min. The PCR products were visualized under UV illumination to ensure the correctness.

### 2.4. Purifying and SNPs Analysis

The PCR products were purified by enzyme, containing 2.5 µL shrimp alkaline phosphatase and 0.05 µL exonuclease I (New England Biolabs, UK), and the purified PCR products were sequenced using ABI PRISM 3730 DNA analyzer (Applied Biosystems, Foster City, CA). The analysis of SNPs was performed on the promoter region of CTLA4, and rs11571315, rs733618, rs4553808, rs11571316, rs62182595, rs16840252, rs5742909, and rs231775 were selected for genotyping.

### 2.5. Statistical Analysis

Statistical analysis was performed through SPSS (SPSS Inc. Released 2008. SPSS Statistics for Windows, Version 17.0. Chicago, USA). The allele frequency of each SNP was analyzed through the exact test to determine whether the SNPs departed from Hardy–Weinberg equilibrium (HWE). The genotype and allele frequencies of the CTLA4 gene were compared between healthy controls and patients with TAARs using the chi-square test or Fisher’s exact test to appraise the association between TAARs and CTLA4 SNPs.

## 3. Results

Nineteen patients and 20 healthy subjects were included. Among 19 patients, there were 4 males and 15 females with a median age of 51 years (range from 2 to 88 years old). The types of TAARs the patients suffered from were allergic reaction (4 cases, 21%) and febrile non-hemolytic transfusion reaction (FNHTR, 15 cases, 79%) (Table 2).

Selected CTLA4 gene SNPs (rs11571315, rs733618, rs4553808, rs11571316, rs62182595, rs16840252, rs5742909, and rs231775) were genotyped in patients and healthy controls. All SNPs were in accordance with the HWE in the control group (*p* > 0.05) (Table 3). Furthermore, there were 4 SNPs (rs4553808, rs62182595, rs16840252, and rs5742909) showing differences in allele frequency between patients with transfusion reaction and healthy controls (Table 3). Among these SNPs, the frequency of “A” allele of rs4553808, “G” allele of rs62182595, “G” allele of rs16840252, and “C” allele of rs5742909 were significantly higher in patients than in controls (0% versus 15%; 0% versus 15%; 0% versus 17.5%; 0% versus 15%, respectively).

Moreover, the “A” allele of rs4553808, the “G” allele of rs62182595, the “G” allele of rs16840252, and the “C” allele of rs5742909 showed significantly higher risk of TAARs (OR = 2.357, 95%CI: 1.584–3.508, *p* = 0.02; OR = 2.357, 95%CI: 1.584–3.508, *p* = 0.02; OR = 2.462, 95%CI: 1.619–3.742, *p* = 0.008; OR = 2.357, 95%CI: 1.584–3.508, *p* = 0.02; OR = 2.357, 95%CI: 1.584–3.508, *p* = 0.02, respectively) (Table 4). The genotype frequency of “CT” genotype in rs11571315 significantly differed between patients with transfusion reaction and healthy controls (Table 4).

## 4. Discussion

According to our results, we demonstrated that 5 SNPs of CTLA4 were correlated with transfusion reactions, and the “A” allele of rs4553808, the “G” allele of rs62182595, the “G” allele of rs16840252, and the “C” allele of rs5742909 showed an increased risk of TAARs. Because these SNPs are in the promoter region of CTLA4, SNP polymorphisms may cause different levels of mRNA transcription, protein translation, and affect T-cell homeostasis [24,25]. Consequently, it suggested that the transcription level of CTLA4 may be related to transfusion reactions. Inhibition of CTLA4 in CD4+CD25+ regulatory T cells (Treg cells) led to impairment of the suppressive function of these cells [26], so CTLA4 gene polymorphism would affect Treg cells and induce TAARs.

A previous study indicated that SNP rs4553808 of CTLA4 is associated with human myasthenia gravis by involvement in transcriptional binding activity for Nuclear Factor I and c/EBPbeta, and G allele of rs4553808 was less frequent in patients with myasthenia gravis than in healthy controls [27]. The genotypes of rs4553808 were all AA and had no G allele in our patient group, so it could be surmised that rs4553808 may participate in the binding of certain transcription factors that cause an immune response to transfused blood components and result in adverse transfusion reactions. The “T” allele of rs5742909 was shown to increase CTLA4 expression and decrease the risk of multiple sclerosis [28] However, the “C” allele of rs5742909 showed an increased risk of TAARs, and it indicated that CTLA4 expression would be altered and cause TAARs. Because rs16840252, rs5742909, and rs4553808 are in strong linkage disequilibrium (LD), the function of rs16840252 could be influenced by rs5742909 or rs4553808 [28]. This might explain why rs16840252 had the greatest effect on the risk of TAARs.

In Table 5, we summarized several studies for clinical conditions of significant CTLA4 SNPs in this study. These SNPs were associated with immune-related diseases or conditions, including Grave’s disease [29], organ or stem cell transplantation [30,31], susceptibility of cancer [32,33,34], and thrombocytopenia [35]. CTLA4 participates in immune regulation and genetic variations in CTLA4 gene would influence immune response and then alter the risk of suffering from a disease [36]. Although the characteristics of allogeneic HSCT and blood transfusion are the same in injecting allogeneic cells into donor, the TAAR-related SNPs which were located in the promoter region in our study were different to those in previous studies for HSCT-related SNPs (rs231775 and rs3087243) [21,37]. rs231775 is a CTLA4 SNP in exon 1 which encodes a signal peptide, and rs231775 abolishes CTLA4 protein expression [37]; rs3087243 is located in the 3′UTR region which regulates mRNA stability [38]. Therefore, mechanisms of CTLA4 influencing HSCT or causing transfusion reactions are different.

The types of TAARs included in the present study were allergic reaction and FNHTR, so these significant CTLA4 SNPs would explain the situation that these adverse reactions occur even when transfusing leukoreduced blood components. Moreover, HPA mismatch between donor and recipient would be associated with FNHTR in patients transfused with leukocyte-poor RBC [13]. Therefore, CTLA4 SNPs and HPA should be further investigated to clarify the mechanisms of FNHTR occurrence in recipients of leukoreduced blood components.

## 5. Conclusions

In conclusion, the present study demonstrated the correlation between CTLA4 gene polymorphism and transfusion reaction, and alleles of 4 CTLA4 SNP with an increased risk of TAARs were found. It is important to explore the potential immune regulatory mechanism affected by SNPs of costimulatory molecules, as this could predict disease occurrence and guide preventive actions.

## Figures and Tables

**Table 1 jcm-08-01961-t001:** CTLA4 primers for promotor and exon 1.

Primers	Sequence
pF	5′ GGCAACAGAGACCCCACCGTT 3′
pR	5′ GAGGACCTTCCTTAAATCTGGAGAG 3′
E1F	5′ CTCTCCAGATTTAAGGAAGGTCCTC 3′
E1R	5′ GGAATACAGAGCCAGCCAAGCC 3′

p: promoter, F: forward primer; R: reverse primer; E1: exon1.

**Table 2 jcm-08-01961-t002:** Characteristics of patients (*n* = 19) and healthy control (*n* = 20).

	Patients, No. (%)	Controls, No. (%)
**Median age of patients**	51 (range, 2–88 years old)	22.8 (range, 22–24 years old)
**Gender**		
Male	4 (21)	5 (20)
Female	15 (79)	15 (80)
**Type of blood transfusion**		
Leukocyte-poor platelet	10 (53)	
Leukocyte-poor RBC	9 (47)	
**Type of transfusion-associated adverse reactions**		
Allergic reaction	4 (21)	
Febrile non-hemolytic transfusion reaction	15 (79)	

**Table 3 jcm-08-01961-t003:** Allele frequencies in patients and controls and odds ratio for transfusion reaction.

SNP	Position	Allele	Minor Allele Frequency		HWE *p* Value	Odds Ratio	*p* Value
			Patient	Control		(95%CI)	
rs11571315	203866178	C/T	0.211	0.300	0.696	1.607 (0.572–4.512)	0.366
rs733618	203866221	T/C	0.474	0.475	0.900	1.228 (0.505–2.988)	0.651
rs4553808	203866282	A/G	0	0.150	0.732	2.118 (1.659–2.703)	0.026 *
rs11571316	203866366	A/G	0.211	0.150	0.628	1.511 (0.470–4.853)	0.486
rs62182595	203866465	A/G	0	0.150	0.732	2.118 (1.659–2.703)	0.026 *
rs16840252	203866796	C/T	0	0.175	0.638	2.152 (1.676–2.762)	0.012 *
rs5742909	203867624	C/T	0	0.150	0.732	2.118 (1.659–2.703)	0.026 *
rs231775	203867991	A/G	0.263	0.300	0.696	1.200 (0.446–3.227)	0.718

HWE: Hardy–Weinberg equilibrium; 95%CI: 95% confidence interval; * *p* < 0.05.

**Table 4 jcm-08-01961-t004:** Genotypes of *CTLA*4 SNPs and their correlations with risk of transfusion reaction.

SNP	Genotype	Genotype Frequency		Odds Ratio	*p* Value
		Patient (*n*)	Control (*n*)	(95%CI)	
rs11571315	CC	3	1	3.563 (0.337–37.687)	0.342
	CT	2	10	0.118 (0.021–0.649)	0.008 *
	TT	14	9	3.422 (0.888–13.183)	0.069
rs733618	CC	6	4	1.846 (0.428–7.962)	0.480
	CT	8	11	0.595 (0.168–2.113)	0.421
	TT	5	5	1.071 (0.254–4.512)	1
rs4553808	AA	19	14	2.357(1.584–3.508)	0.020 *
	AG	0	6	0.424(0.285–0.631)	0.020 *
	GG	0	0	NA	NA
rs11571316	GG	14	15	0.933 (0.222–3.930)	0.342
	AG	2	4	0.471 (0.086–2.932)	1
	AA	3	1	3.563 (0.337–37.687)	0.648
rs62182595	GG	19	14	2.357 (1.584–3.508)	0.020 *
	AG	0	6	0.424 (0.285–0.631)	0.020 *
	AA	0	0	NA	NA
rs16840252	CC	19	13	2.462 (1.619–3.742)	0.008 *
	CT	0	7	0.406 (0.267–0.618)	0.008 *
	TT	0	0	NA	NA
rs5742909	CC	19	14	2.357 (1.584–3.508)	0.020 *
	CT	0	6	0.424 (0.25–0.631)	0.020 *
	TT	0	0	NA	NA
rs231775	GG	12	9	2.095 (0.581–7.555)	0.256
	AG	4	10	0.267 (0.065–1.091)	0.062
	AA	3	1	3.563 (0.337–37.687)	0.342

95%CI: 95% confidence interval; *: *p* < 0.05; NA: not applicable.

**Table 5 jcm-08-01961-t005:** Summary of *CTLA*4 SNPs included and significant in present study and related clinical conditions.

SNP	Clinical Condition	Reference
rs5742909	Grave’s disease	[29]
	long-term kidney allograft function	[30]
	cancer predisposition	[32]
rs4553808	viral infection in kidney transplantation	[31]
	long-term kidney allograft function	[30]
	cancer predisposition	[32]
rs16840252	colorectal cancer	[33]
	gastric adenocarcinoma	[34]
rs11571315	immune thrombocytopenia	[35]

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
