# Peer review of "Association of CTLA4 Gene Polymorphism with Transfusion Reaction after Infusion of Leukoreduced Blood Component"

_jcm, 2019, doi:10.3390/jcm8111961_

Round 1

Reviewer 1 Report

This short report describes the association of CTLA4 gene polymorphims with adverse transfusion reaction. The paper is interesting and well written. Suggestions for improvements are;

1- page 1, line 38; fix "components.2"

2-Table 2: an additional column should be inserted on the right to present demographic data related to the healthy controls.

3- page 6, line 160; change "hematopoietic stem cell transplantation (HSCT)" for "HSCT, as this abbreviation was already defined beforehand.

END

Author Response

Reviewer 1

This short report describes the association of CTLA4 gene polymorphims with adverse transfusion reaction. The paper is interesting and well written.

Author reply: We appreciated the reviewer’s comment.

page 1, line 38; fix "components.2"

Author reply: We have revised them.

Table 2 an additional column should be inserted on the right to present demographic data related to the healthy controls.

Author reply: We have added demographic data related to the healthy controls.

page 6, line 160; change "hematopoietic stem cell transplantation (HSCT)" for "HSCT, as this abbreviation was already defined beforehand.

  Author reply: We have revised them.

Reviewer 2 Report

In this interesting study, the authors have examined the CTLA4 gene polymorphism in patients with transfusion-associated adverse reactions (most of which are Febrile non-hemolytic transfusion reaction), and found that 4 CTLA4 SNP ("A" allele of rs4553808, "G" allele of rs62182595, "G" allele of rs16840252, and "C" allele of rs5742909) were correlated with an increased risk of TAARs.

Major comments:

The numbers of patients and healthy control are low in this report. Could the authors show more patients and healthy control?

Do the patients and healthy subjects show a significantly different in ethnicity?

The ethnicity information is important to know for comparing the CTLA4 gene polymorphism Could the authors discuss the reasonable mechanism of correlation of CTLA4 gene polymorphism with febrile non-hemolytic transfusion reaction, which is mostly induced by cytokines and HLA antigens in blood unit?

Methodology

Please check the writing of number 2 in line 38.

Author Response

Reviewer 2

In this interesting study, the authors have examined the CTLA4 gene polymorphism in patients with transfusion-associated adverse reactions (most of which are Febrile non-hemolytic transfusion reaction), and found that 4 CTLA4 SNP ("A" allele of rs4553808, "G" allele of rs62182595, "G" allele of rs16840252, and "C" allele of rs5742909) were correlated with an increased risk of TAARs.

Author reply: We appreciated the reviewer’s comment.

The numbers of patients and healthy control are low in this report. Could the authors show more patients and healthy control?

  Author reply: TAARs rarely occur after leukoreduced blood products transfusion, and Linkou Chang Gung Memorial Hospital is a tertiary medical center with nearly 4000 beds. Therefore, the data was very precious.

Do the patients and healthy subjects show a significantly different in ethnicity?

  Author reply: They are all Taiwanese.

Could the authors discuss the reasonable mechanism of correlation of CTLA4 gene polymorphism with febrile non-hemolytic transfusion reaction, which is mostly induced by cytokines and HLA antigens in blood unit?

  Author reply: We have discussed reasonable mechanism of correlation of CTLA4 gene polymorphism with febrile non-hemolytic transfusion reaction in the first paragraph of Discussion (line 142-144) as follows: “Inhibition of CTLA4 in CD4+CD25+ regulatory T cells (Treg cells) led to impairment of the suppressive function of these cells [26], so CTLA4 gene polymorphism would affect Treg cells and induce TAARs.”

Please check the writing of number 2 in line 38

  Author reply: We have revised them.

Reviewer 3 Report

The article is well written overall, with proper references.

"The authors investigated relationship between CTLA4 polymorphism and rejection of blood transfusion. Authors genotyped selected CTLA4 gene SNPs and compared between case (people with transfusion rejection) and control. The authors were able to demonstrate a statistically significant presence of CTLA4 mutation in cases, as compared to controls.

Information like this can enable better screening of donors, in order to prevent transfusion-associated adverse reactions. Since blood transfusion is a critical and very common practice in medicine, this article has a broad base of interest. The writing quality is good. Experiments are well designed. Results and ensuing conclusion flow well.

The overall novelty is average, since there are some evidence of relationship between CTLA4 and adverse transplantation effects in the literature

Author Response

Reviewer 3

The article is well written overall, with proper references. "The authors investigated relationship between CTLA4 polymorphism and rejection of blood transfusion. Authors genotyped selected CTLA4 gene SNPs and compared between case (people with transfusion rejection) and control. The authors were able to demonstrate a statistically significant presence of CTLA4 mutation in cases, as compared to controls. Information like this can enable better screening of donors, in order to prevent transfusion-associated adverse reactions. Since blood transfusion is a critical and very common practice in medicine, this article has a broad base of interest. The writing quality is good. Experiments are well designed. Results and ensuing conclusion flow well. The overall novelty is average, since there are some evidence of relationship between CTLA4 and adverse transplantation effects in the literature

Author reply: We appreciated the reviewer’s comment.